# Leg Muscle Activity and Perception of Effort before and after Four Short Sessions of Submaximal Eccentric Cycling

**DOI:** 10.3390/ijerph17217702

**Published:** 2020-10-22

**Authors:** Pierre Clos, Romuald Lepers

**Affiliations:** INSERM UMR1093-CAPS, Université Bourgogne Franche-Comté, UFR des Sciences du Sport, F-21000 Dijon, France; romuald.lepers@u-bourgogne.fr

**Keywords:** semi-recumbent cycling, electromyography, familiarization, interindividual variability

## Abstract

*Background*: This study tested muscle activity (EMG) and perception of effort in eccentric (ECC) and concentric (CON) cycling before and after four sessions of both. *Methods*: Twelve volunteers naïve to ECC cycling attended the laboratory six times. On day 1, they performed a CON cycling peak power output (PPO) test. They then carried-out four sessions comprising two sets of 1 to 1.5-min cycling bouts at 5 intensities (30, 45, 60, 75, and 90% PPO) in ECC and CON cycling. On day 2 and day 6 (two weeks apart), EMG root mean square of the vastus lateralis (VL), rectus femoris (RF), biceps femoris (BF), and soleus (SOL) muscles, was averaged from 15 to 30 s within each 1-min bout and perception of effort was asked after 45 s. *Results*: Before the four cycling sessions, while VL EMG was lower in ECC than CON cycling, most variables were not different. Afterwards, ECC cycling exhibited lower RF EMG at 75 and 90% PPO (all *p* < 0.02), lower VL and BF EMG at all exercise intensities (all *p* < 0.02), and inferior SOL EMG (all *p* < 0.04) except at 45% PPO (*p* = 0.07). Perception of effort was lower in ECC cycling at all exercise intensities (all *p* < 0.03) but 60% PPO (*p* = 0.11). *Conclusions*: After four short sessions of ECC cycling, the activity of four leg muscles and perception of effort became lower in ECC than in CON cycling at most of five power outputs, while they were similar before.

## 1. Introduction

In contrast to the type of pedaling we all know (i.e., concentric (CON) cycling), eccentric (ECC) cycling makes the muscle lengthen while contracting. Mostly implemented on semi-recumbent bicycles, it consists in resisting the force of the pedals that are driven backward by an engine. Submaximal ECC cycling is used as a rehabilitation tool because of its low cardiorespiratory demand associated with a relatively high force output [1,2,3,4,5,6]. When matched for power output, 16 sessions of CON or ECC cycling improved maximal voluntary force similarly, but ECC cycling was perceived easier [7].

As ECC cycling is often utterly new to individuals, most studies familiarized their participants with it, which refines their motor control and translates into lower muscle activity (EMG) [8]. In Peñailillo et al., (2013), volunteers merely pedaled for 5 min at a power output of 50 W, whereas in Lechauve et al., (2014) they completed 6 sessions lasting 5 to 15 min at an intensity ranging from 50 W to 300 W. Despite these differences, the EMG (root mean square, RMS) of all leg muscles studied was systematically lower in ECC than CON cycling at the same power output: this was the case for the vastus lateralis muscle (VL) [9,10,11,12], the rectus femoris (RF), and the biceps femoris (BF) muscles [11,12], and for the vastus medialis muscle [12]. These studies used single exercise intensities ranging from 65 to 85% CON cycling peak power output (PPO).

Only LaStayo et al., (2008) [13] addressed the changes in leg muscle EMG following several sessions of submaximal ECC cycling, observing a decrease in VL muscle EMG RMS during ECC cycling at a power output of 200 W, after eight weeks of training.

The present investigation aimed to follow-up on aspects of the study of Lastayo et al. (2008), by comparing the EMG RMS in CON with ECC cycling. However, given the novelty of ECC cycling, we performed measurements while participants were not familiar with the task (they had pedaled for 3 min only), and after four short sessions of ECC and CON cycling (comprising 10 to 15 min of both). In order to provide a more comprehensive view of the changes observed, pedaling intensities ranged from 30 to 90% PPO and the activity of four leg muscles was assessed. ECC cycling also elicits particular perceptual responses [14], the gap in EMG RMS at a given mechanical power output is accompanied by a perception of less effort in ECC than in CON cycling [13,15,16,17]. The second aim of this study was therefore to assess whether the four cycling sessions would affect perception of effort.

Based on preliminary results, we expected that the typically observed lower muscle activity during ECC than CON contraction at a given power output would be counterbalanced by factors related to the novelty of the task. Consequently, we hypothesized that EMG RMS of the four leg muscles studied would not differ between ECC and CON cycling before the cycling sessions, but that it would be lower in ECC than CON cycling afterwards. Changes in perception of effort would mirror changes in muscle activity given the strong correlation between the two parameters [18].

## 2. Material and Methods

### 2.1. Participants

Based on the difference in EMG reported by Lastayo et al., (2008) before training, a required sample size of 10 was calculated for a one-tail matched pairs test, an effect size of d = 0.9, an alpha level of 0.05 and power of 0.8.

Twelve healthy leisurely active volunteers, naïve to ECC cycling (8 men and 4 women; 24 ± 2 years; 174 ± 8 cm; 67 ± 11 kg) signed an informed consent form. The study was conducted in accordance with the declaration of Helsinki (2008).

### 2.2. Experimental Approach

Participants attended the laboratory on 6 different days (Figure 1). Two semi-recumbent cycle ergometers were used for steady-power CON (Ergoline GmbH, Lidenstrabe, Germany) and ECC (Cyclus 2, Cyclus GmbH, Leipzig, Germany) cycling. The knee and hip joint flexion angles were measured at the maximal knee extension possible on each bicycle, and the seat seatback and incline were adjusted so that there was less than 10° of difference between either of the two joint angles on the two bikes.

On the first day, volunteers performed a maximal incremental CON cycling test at 60 rpm (starting at 50 W with an increase of 1 W every 3 s). Power output (PPO) and heart rate (bpm) were recorded at exhaustion (i.e., when pedaling cadence dropped below 50 rpm for more than 10 s). The maximal test was followed by 3 min of ECC cycling at 40% PPO to introduce participants to the task. They then carried out a total of 50 min of both ECC and CON cycling in four sessions spread over two weeks. The first session took place a week after the incremental test, and a PRE-test was integrated in it. The second session was three days later, and the two other sessions were conducted the next week on the same days. A POST-test was carried-out on the sixth visit, two weeks after the PRE-test.

The first and second sessions consisted in pedaling 1 min twice at five different exercise intensities (30, 45, 60, 75, and 90% of PPO) on both bicycles, and recovered between bouts until heart rate had returned to baseline level ±10 bpm. Bouts of the third and fourth sessions lasted 1.5 min. On the sixth day, participants pedaled for 1 min at each power output on each bicycle. Power outputs and exercise modalities were performed in random order so as to avoid the confound of an order effect. Participants were asked to pedal at a cadence of 60 rpm and were oblivious to any objective marker of exercise intensity (i.e., power output or heart rate), across all sessions.

Physiological and perceptual measurements (Figure 1) were carried out during the PRE- and POST-tests, after a warm-up consisting of 2.5 min of CON and ECC cycling at 40% of PPO.

Participants were asked to keep their usual diet and physical activities, but to restrain from any intense physical activity for two days before the testing days.

Since the knee extensor, the hip extensor, and the ankle plantar flexor muscles are all heavily involved during ECC cycling [19], we recorded the EMG of the RF, the VL, the BF, and the soleus (SOL) muscles.

### 2.3. Physiological Measurements

Heart rate was reported 45 s after the onset of each cycling bout using a chest-belt heart-rate monitor (Ergoline GmbH, Lidenstrabe, Germany).

EMG was measured using 2 square surface Ag/AgCl electrodes (10 mm^2^) touching each other in a bipolar setting, with a reference electrode on the right patella. The skin was first shaved and cleaned with alcohol swabs. Electrodes were identically placed on the belly of four muscles using anatomical landmarks (seniam.org); the VL, RF, BF, and SOL muscles.

The EMG signal was amplified by a thousand (2 kHz) using Acqknowledge 5.0 software linked to an MP160 unit (Bipoac Systems Inc., Santa Barbara, CA, USA). Based on the frequency distribution, the signal was band-pass filtered from 20 to 380 Hz, and a 1-Hz wide notch filter was applied every 50 Hz, using a second order Butterworth filter.

### 2.4. Perceptual Measurements

Perception of effort (“conscious sensation of how hard, heavy, and strenuous the task is” [20], was obtained by asking the volunteers the difficulty they felt to breathe and to drive their legs while pedaling [21]. Perception of the difficulty to pedal at the requested cadence corresponded to the strain the volunteer experienced to stay at 60 ± 2 rpm, not taking overall perception of effort into account. The two perceptions were reported during the last 10 s of each cycling bout using Borg’s CR100 scale [22], with which participants were familiarized during the incremental test. Participants were instructed to disregard pain or discomfort when reporting perceived effort, and memory anchoring was used for perception of effort (a rating of 100 on the CR100 scale corresponded to the most intense effort they had experienced during an endurance exercise).

### 2.5. Data Analysis

EMG was analyzed from 15 s to 45 s of each bout (Figure 1), to ensure that the participants had sufficient time to focus on pedaling and that their concentration was not interrupted by the experimenter asking for their perceptions after 50 s. EMG RMS was averaged over this 30 s period for each muscle, exercise intensity and day, and normalized to the EMG RMS in CON cycling at 30% PPO measured the same day (PRE or POST). Heart rate was expressed as a percentage of the peak value reached during the incremental test. The interindividual coefficient of variation, corresponding to the standard deviation relative to the mean (%), was calculated for each dependent variable. The difference between each variable in CON and ECC cycling at all power outputs was averaged for PRE- and POST-test (see Figure 2 for EMG data), as was the interindividual coefficient of variation of each variable.

### 2.6. Statistical Analyses

All data are expressed as mean ± standard deviation, and alpha level for significance was set at *p* < 0.05. All data of an array that were more than 2 standard deviations below or above the mean value were removed (7.3% of the data).

Variables that were not normally distributed according to the Shapiro–Wilk test (EMG RMS; heart rate; perceptions; difference between the EMG RMS of the VL and SOL muscles in CON and ECC cycling) were analyzed using Friedman’s non-parametric ANOVA. A Wilcoxon matched-pairs test was conducted for any significant ANOVA result. These data were compared in ECC vs. CON cycling at PRE and POST. The average difference between CON and ECC cycling values at all exercise intensities was analyzed using a repeated-measure one-way ANOVA for each variable, with TIME (PRE, POST) as factor. The average interindividual coefficient of variation of all exercise intensities was tested with a two-way repeated measure ANOVA with TIME (PRE, POST) and MODALITY (ECC, CON) as factors. Cohen’s dz (2013) [23] were calculated for non-parametric and follow-up analyses (G*Power software version 3.1.9.4; Kiel University, Kiel, Germany).

## 3. Results

At the end of the incremental CON cycling test, participants attained a heart rate of 179 ± 10 bpm, for a PPO of 244 ± 44 W, and a perception of effort of 89 ± 6 a.u.

### 3.1. Muscle Activity

As shown in Figure 2, the average individual EMG RMS of the RF muscle went from greater in ECC at PRE to greater in CON cycling at POST (−28.5 ± 88 vs. 45.9 ± 59.2% points; *p* < 0.01; $\eta_{p\;}^{2}$ > 0.53). The VL, BF, and SOL muscles showed no significant effect of time (59.4 ± 66.9 vs. 63.4 ± 56.4% points; all *p* > 0.46; all $\eta_{p\;}^{2}$ < 0.56). Figure 3 shows the EMG RMS of all muscles before and after the four cycling sessions and Table 1 gives the effect sizes (dz) of the EMG RMS comparisons between CON and ECC cycling at PRE and POST.

### 3.2. Heart Rate

Figure 4 shows heart rate together with perceptual responses. At PRE, heart rate was lower in ECC than CON cycling at all exercise intensities (all *p* < 0.04; all dz > 0.83) except at 30% PPO (*p* = 0.17; dz = 0.46), while it was higher at all power outputs at POST (all *p* < 0.001; all dz > 1.8).

The average difference between heart rate in ECC and CON cycling at all exercise intensities did not change from PRE to POST (*p* = 0.09; $\eta_{p\;}^{2}$ = 0.3).

### 3.3. Perceptual Responses

At PRE, perception of effort (Figure 4) was greater in ECC than CON cycling at 30% PPO (*p* = 0.03; dz = 0.56), but similar from 45 to 75% PPO (all *p* > 0.13; all dz < 0.23), and lower in ECC at 90% PPO (*p* = 0.03; dz = 0.7). At POST, effort was lower in ECC at all exercise intensities (all < 0.03; all dz > 0.52) except 60% PPO (*p* = 0.11; dz = 0.52).

Before the cycling sessions, perception of the difficulty to pedal at the instructed cadence was higher in ECC at 60% and 75% PPO (all *p* < 0.05; all dz > 0.49), while it was similar at all power outputs afterwards (all *p* > 0.26; all dz < 0.45).

The average difference between perception of effort at all exercise intensities in CON and ECC cycling did not change from PRE to POST (*p* = 0.3; $\eta_{p\;}^{2}$ = 0.11), whereas the difference in perception of the difficulty to pedal at the instructed cadence decreased (−5.5 ± 6.8 vs. 0.2 ± 3.2 a.u; *p* = 0.02; $\eta_{p\;}^{2}$ = 0.48), bridging the coordination gap between the exercise modalities.

### 3.4. Heterogeneity of the Responses

Pooling the PRE- and POST-tests, the average inter-individual coefficient of variation of all intensities (Table 1) was greater in ECC than CON cycling for heart rate, the EMG RMS of all muscles (Figure 2) and perception of effort (all *p* < 0.05; all $\eta_{p\;}^{2}$ > 0.46). Following the cycling sessions, the average inter-individual coefficient of variation declined in both modalities for heart rate, RF EMG, BF EMG RMS, and SOL EMG RMS (all *p* < 0.03; all $\eta_{p\;}^{2}$> 0.52).

Heart rate inter-individual coefficient of variation decreased from PRE to POST in CON (*p* < 0.001; dz = 1.26) and ECC cycling (*p* < 0.001; dz = 5). SOL EMG RMS inter-individual coefficient of variation dropped in both CON (*p* = 0.02; dz = 0.36) and ECC cycling (*p* < 0.001; dz = 1.64), and that of BF EMG RMS declined in CON only (*p* > 0.001; dz = 7.7) but not in ECC cycling (*p* = 0.49; dz = 0.5).

## 4. Discussion

We investigated whether four sessions of 10 to 15 min of submaximal ECC cycling would affect the activity of four leg muscles and perception of effort over a range of submaximal exercise intensities. Before the intervention, the activity of the RF, BF, and SOL muscles was not different between CON and ECC cycling at most exercise intensities, while afterwards, the RF muscle was less activated in ECC than CON cycling at the two highest exercise intensities, the VL and the BF muscles were less activated in ECC cycling at all exercise intensities, and the SOL muscle was less activated in ECC cycling at four of five exercise intensities.

Before the cycling sessions, effort was perceived mostly similar during ECC and CON cycling, whereas afterwards it was globally lower during ECC cycling.

### 4.1. Changes in Leg Muscle Activity

After, but not before, the cycling sessions, leg muscle activity was globally lower during ECC than CON cycling. This may be due to specific muscle adaptations that occur through repeated bouts of ECC contractions. This “repeated-bout effect” is observed after completion of a single ECC session [24] and seems to be accompanied by improved synchronization of the motor units, the addition of sarcomeres in series [25], enhanced excitation-contraction coupling [26], and increased muscle stiffness [27]. All these phenomena are assumed to foster neuromuscular efficiency, so that a lower motor command—reflected by muscle activity [28]—is required to develop the same torque as before the ECC session(s). These adaptations may have occurred in all the muscles we studied. Each muscle, however, showed distinct changes in muscle activity, in relation to its specific contribution to pedaling. However, the very limited training stress (short duration and long recovery periods) let think that these mechanisms were not predominant, and that a more refined motor control is more likely to have caused the decrease in EMG. This is claim is in line with a lower difficulty to pedal at the instructed cadence (see Section 4.2).

The already lower VL muscle activity in ECC than CON cycling (except at 45% PPO) while participants were unfamiliar with ECC cycling might be due to the fact that the VL muscle has for main role to adjust the amount of force produced by the quadriceps muscles [29]; the nature of its function would not change with practice. In LaStayo et al., (2008), an eight-week ECC cycling training program halved the activation time of the VL muscle within a pedaling cycle, which the authors suggested was the result of motor learning. We should nonetheless note that their participants were totally naïve to ECC cycling when tested before the training period, and our results let think that the changes they reported would probably have been less important if the participants had been introduced to the task beforehand. In any case, it seems that a four-session cycling period only marginally affected the activity of the VL muscle.

The activity of the RF muscle of our participants became lower in ECC than CON cycling at high power outputs only (i.e., 75 and 90% PPO). Nonetheless, the four cycling sessions period increased the average difference between the muscle activity of the RF in the two exercise modalities, showing that this muscle was generally recruited less in ECC cycling after participants had performed four short sessions of it. Since the activation of the RF muscle during knee extension only intensifies at higher loads [29], it may play a similar role in CON and ECC cycling up to 60% PPO. Its greater activity in CON cycling at higher power outputs would support the agonist muscles in developing power. As the RF muscle serves as a hip flexor as well as a knee extensor, when utterly naïve to ECC cycling, individuals might have used it to pull the pedal, thereby generating negative torques and counterbalancing the work of agonist muscles (i.e., the knee extensors and hip flexors)–increasing its overall activity.

The BF muscle exhibited the greatest change in muscle activity, from being lower in ECC than CON cycling only at 75% PPO before the four cycling sessions, to lower at all exercise intensities afterwards. It is possible that the BF muscle was less co-activated when the knee extensors were resisting the pedals [11].

The muscle activity of the SOL muscle was only lower in ECC than CON cycling at 30 and 90% at PRE, while it was lower at all exercise intensities but 45% PPO in at POST, with large to huge effect sizes. Among the ankle dorsi-flexors, we focused on the SOL muscle suspecting it would be highly involved when unfamiliar with ECC pedaling, providing information about the task through its high number of afferent fibers [30]. We thus speculate that its lower activity after the four cycling sessions could partly be accounted for by less output from the Ia afferent fibers via two underlying mechanisms. First, repeating ECC sessions may have decreased the H-reflex [31]. Second, as shown by Peñailillo et al., (2015) in a second bout of eccentric cycling, muscle fascicles stretch less at a given power output.

Finally, the overall lower muscle activity in ECC cycling following the four cycling sessions may partly be the result of an enhanced activity of central pattern generators at the spinal level [32]. Nevertheless, because during ECC exercises the excitability of the spinal tract is very low [33], and brain activity is high and lasts longer than during CON contraction [15,34], there may be more margin for improvement at the cortical level.

### 4.2. Changes in Perceptual Responses and Heart Rate

Perception of effort is thought to be the result of a copy of the motor command (or corollary discharge) towards somatosensory areas of the brain [21,35]. Since muscle activity is believed to reflect central motor command [28], we expected the lower leg muscle activity in ECC after the four cycling sessions to be associated with a lower perception of effort [18].

At PRE, the greater perception of effort in ECC than CON cycling at 30% PPO, together with the opposite finding at 90% PPO and the absence of difference in between these power outputs (Figure 4), seem to highlight a shift in the mechanisms responsible for perception of effort as exercise intensity increased. Namely, at 90% during ECC cycling, the intensity of the central comand—indirectly generating perceived effort—would be predominantly determined by exercise intensity (i.e., applying greater torque to the pedals), while at 30% it would result in a greater proportion from the motor control required by the task. This is in line with the fact that the difference in the average perception of the difficulty to pedal at the instructed cadence in CON and ECC cycling disappeared after the four cycling sessions. It thus seems that the lower perception of effort in ECC than in CON cycling after (but not before) the cycling sessions also partly originates from less difficulty pedaling at the required cadence. This might be the result of more refined motor coordination, accounting for lower muscle activity [13]. During ECC cycling, perception of effort seems to be more due to coordination than to the production of power, while mechanisms seem reversed during CON cycling.

The lower heart rate after the four cycling sessions would indicate that the energy demand was lower in ECC cycling, requiring a weaker motor command, in turn leading to the perception of less effort. This suggestion is consistent with the lower muscle activity of the BF muscle at 30% PPO only at POST (and not at PRE) and increased effect size in the difference in EMG RMS between CON and ECC cycling of the other muscles. Nonetheless, caution is required when using heart rate as a surrogate for metabolic intensity, because its change during an ECC exercise does not follow that of oxygen consumption as closely as it does in a CON exercise [36].

### 4.3. Variability of the Responses

Both physiological and perceptual responses exhibited greater inter-individual variability in ECC than in CON cycling, and the cycling sessions reduced the heterogeneity of all physiological parameters in the two exercise modalities (Table 2 and Figure 2). Heart rate and SOL muscle activity became more homogeneous (lower interindividual coefficient of variation) in both modalities, although effect sizes indicate a larger effect in ECC. On the other hand, the variability in BF muscle activity diminished in CON cycling only. Overall, these results suggest that a four-session familiarization period with ECC cycling can reduce the influence of the novelty of the task on individual responses, and help understanding the effects of the task per se. Unexpectedly, we also observed reduced variability during CON cycling—possibly related to the use of semi-recumbent rather than standard bicycles. The four cycling sessions eliminated variability that is inherent not to the task but to its novelty, even in supposedly familiar tasks.

### 4.4. Limits and Perspectives

In addition to several inherent limits of surface EMG [37], dynamic surface EMG measurements are affected by factors such as changes “in the geometry of the electrode-muscle system”, which modulate the amplitude of the EMG signal in addition to the actual level of muscle activity [38]. Nonetheless, as the pedaling rate did not vary and EMG RMS was calculated over 30 pedaling cycles, this limitation probably did not influence the changes in EMG RMS data significantly. Furthermore, combining joint angle and EMG signal would have been of great value in order to analyze EMG throughout the pedaling cycle. Finally, in order to determine the optimal duration for a familiarization procedure with submaximal eccentric cycling, it would be of great interest to apply a method similar to that of Green et al., (2017) [39], who assessed the between-session reliability in leg muscle activity when performing 10 s ECC cycling sprints.

## 5. Conclusions

Four short cycling sessions of submaximal ECC cycling led to lower activities of the RF, VL, BF, and SOL muscles, and perception of less effort in ECC than CON cycling over a range of power outputs—while most variables were not different between exercise modalities beforehand. The inter-individual differences in muscle activity and heart rate declined after the cycling sessions but were overall greater in ECC cycling. Future investigations should consider implementing several sessions of ECC cycling before carrying-out any muscle activity or perceptual assessments.

## Figures and Tables

**Figure 1 ijerph-17-07702-f001:**
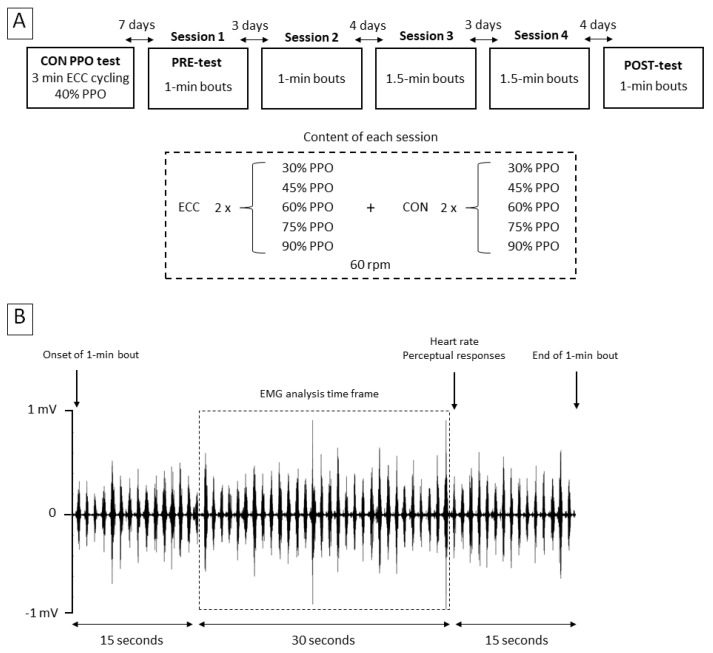
Overview of the protocol. In panel A, each of the six squares represents a session in the laboratory. Measurements were carried out in eccentric and concentric cycling during the first set of the first session (PRE-test) and during the sixth session (POST-test). EMG of the rectus femoris, vastus lateralis, biceps femoris, and soleus muscles, as well as perceptions of effort and of the difficulty to pedal at the instructed cadence (60 rpm), were recorded. Panel B shows an example of a raw and unfiltered EMG signal (at PRE during CON cycling at 60% PPO), with the timing of physiological (EMG and heart rate) and perceptual measurements (perceived effort and difficulty to pedal at the instructed cadence). PPO: Peak power output; CON: concentric cycling; ECC: eccentric cycling.

**Figure 2 ijerph-17-07702-f002:**
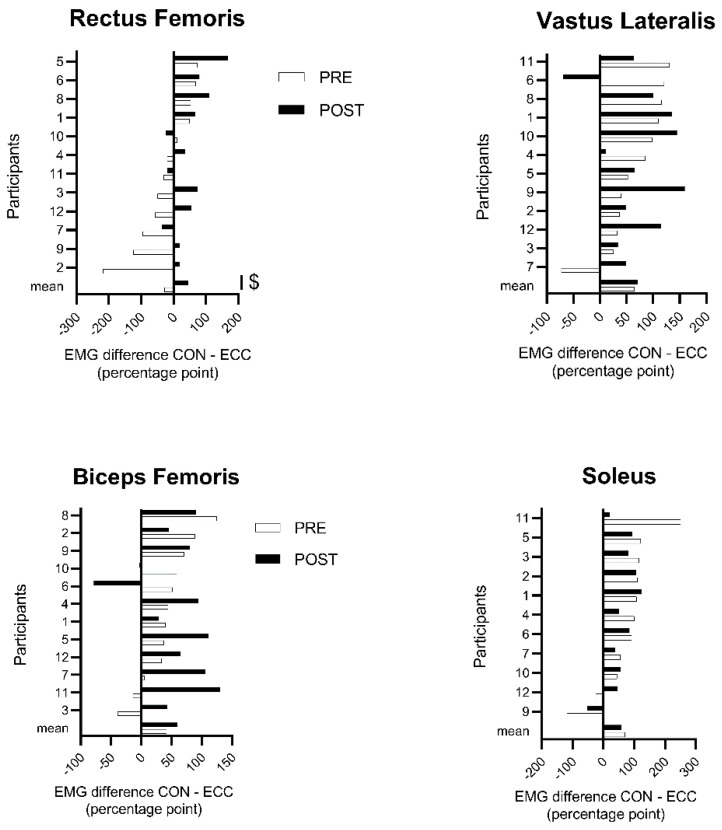
Individual difference in EMG between concentric and eccentric cycling. This figure shows the difference (percentage point) in the root mean square of the EMG between CON and ECC cycling, averaged for all exercise intensities per muscle and per individual, before (PRE) and after (POST) the four sessions of cycling. Note that the scale of the abscissa differs for each chart. A negative score means that EMG RMS was greater in ECC than CON cycling. ^$^: Difference between PRE and POST at a given power output (*p* < 0.05). Error bars represent the standard error.

**Figure 3 ijerph-17-07702-f003:**
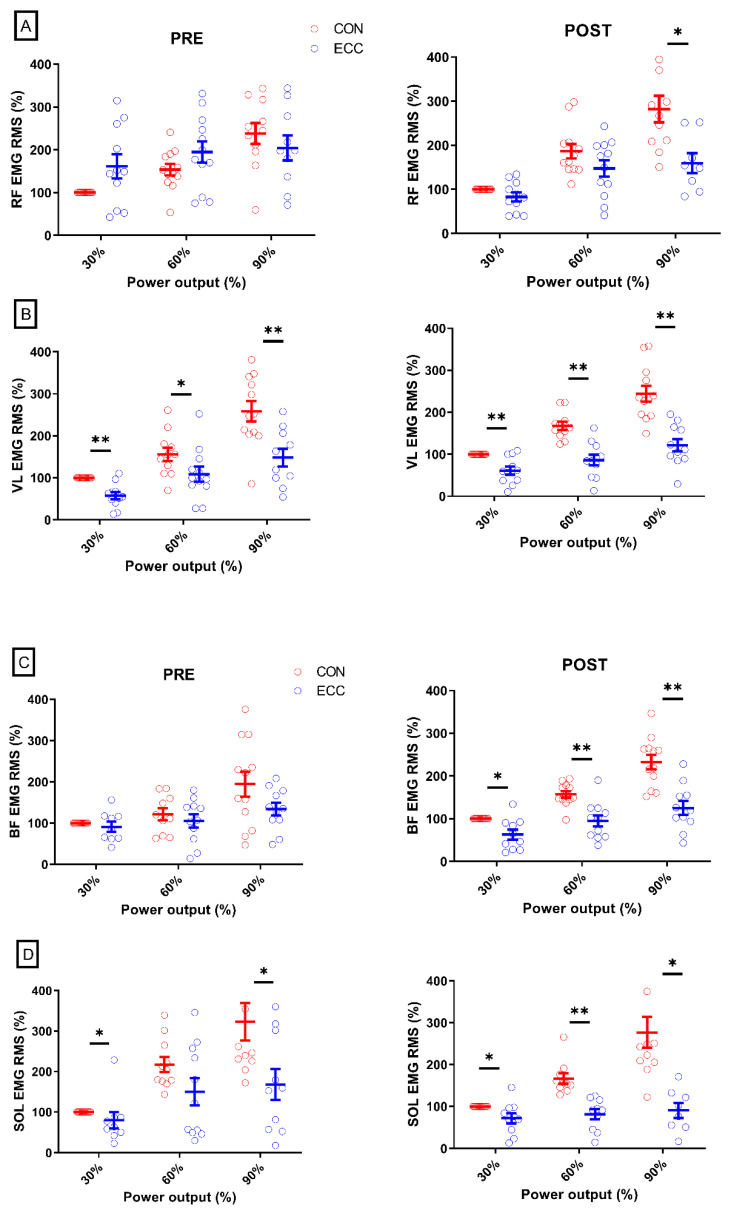
Muscle activity. Panel **A** displays the scattered muscle activity (EMG RMS) of the rectus femoris (RF) muscle before (PRE) and after (POST) the four sessions of cycling. Panels **B**–**D** show the EMG RMS of the vastus lateralis (VL), biceps femoris (BF), and soleus (SOL) muscles, respectively. Data were normalized to the EMG RMS in CON cycling at 30% of the peak power output the same day. Only three exercise intensities are displayed for sake of clarity. Difference between CON and ECC cycling value at a given power output: * means *p* < 0.05; ** means *p* < 0.01. Error bars represent the standard error.

**Figure 4 ijerph-17-07702-f004:**
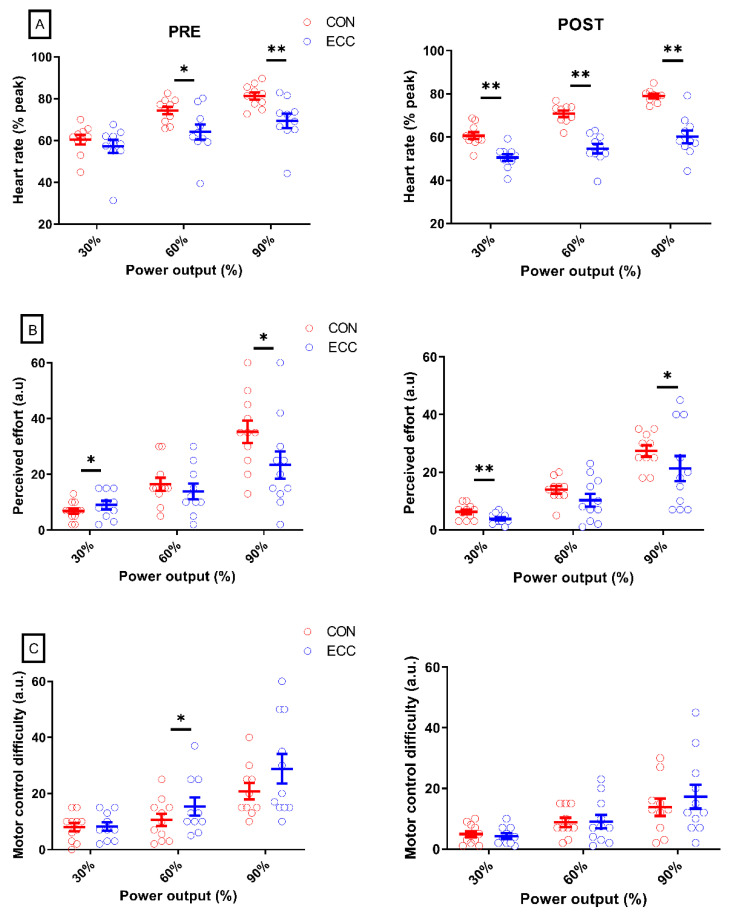
Heart rate and perceptual responses. Panel **A** represents the heart rate before (PRE) and after (POST) the four sessions of cycling. Panel **B** shows perceived effort, and panel **C** the perceived difficulty to pedal at the requested cadence (“motor control difficulty”). Only three exercise intensities are displayed for sake of clarity. Difference between CON and ECC cycling values at a given power output: * means *p* < 0.05; ** means *p* < 0.01. Error bars represent the standard error.

**Table 1 ijerph-17-07702-t001:** Intraindividual effect sizes for muscle activity before and after the cycling sessions.

Variable	Intensity (% PPO)	Effect Size PRE (dz)	Effect Size POST (dz)
EMG RMS of Rectus femoris	45	−0.45	0.33
	60	−0.46	0.57
	75	−0.05	1.97
	90	0.51	1.29
EMG RMS of Vastus lateralis	45	0.44	1.17
	60	0.54	1.57
	75	1.28	1.42
	90	0.97	1.56
EMG RMS of Biceps femoris	45	0.47	0.82
	60	0.43	1.32
	75	0.65	1.2
	90	0.6	1.66
EMG RMS of Soleus	45	1.65	0.78
	60	0.63	1.51
	75	0.6	1.84
	90	1.07	2.31

The sign “−” signifies that the mean in concentric cycling was lower than that in eccentric cycling.

**Table 2 ijerph-17-07702-t002:** Inter-individual coefficients of variation of all variables.

	PRE		POST	
	ECC	CON	ECC	CON
Perceived effort ^$$^ (%)	58.5 ± 12.5	47.2 ± 8	58.9 ± 14.4	30.3 ± 7.3
Perceived difficulty to match cadence (%)	63.2 ± 5.5	61.1 ± 10.4	71.8 ± 6.4	63.4 ± 6.6
Heart rate ^$$$,###^(%)	20.2 ± 1.5	13.8 ± 1.1	10.3 ± 0.8 ***	5.9 ± 0.8 ***
RF EMG RMS ^$$,#^(%)	51.3 ± 7.4	31.2 ± 3.7	43.6 ± 3.3	28.8 ± 7.1
VL EMG RMS ^$$$^ (%)	56.1 ± 10	26.7 ± 7.6	47.3 ± 5.6	25.8 ± 10.1
BF EMG RMS ^$$,###^ (%)	54.6 ± 9.9	44.3 ± 9.1	50.3 ± 7.1	18.1 ± 6.7 ***
SOL EMG RMS ^$$,###^ (%)	67.5 ± 6.9	33.2 ± 13.8	52.1 ± 3.5 ***	31.3 ± 10.5 *

The inter-individual coefficients of variation of all exercise intensities are averaged for each variable. VL: vastus lateralis muscle; RF: rectus femoris muscle; BF: biceps femoris muscle; SOL: soleus muscle; RMS: root mean squared. Greater value in ECC than CON without interaction effect (^$$^ means *p* < 0.001; ^$$$^ means *p* < 0.001). Decline from PRE and POST without interaction effect (^#^ means *p* < 0.05; ^###^ means *p* < 0.001). Decrease from PRE to POST in ECC or CON (* means *p* < 0.05; *** means *p* < 0.001).

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
