# Peer review of "Leg Muscle Activity and Perception of Effort before and after Four Short Sessions of Submaximal Eccentric Cycling"

_ijerph, 2020, doi:10.3390/ijerph17217702_

Round 1

Reviewer 1 Report

First of all, this study is very interesting and provides a novel information, due to the lack of this type of experiments.

I have a few suggestions to the authors:

(1). In the introduction section, explain with more detaisl the rationale o ECC cycling and waht is the main difference fomr cycling

(2). Provide a sample size calculation and power of tests;

(3). Reorganize section 2.2, add the text before the figure;

(4). I think that a section to EMG analysis should include the fig B;

(5). Divide section 2.3 into more section to make manuscript easier to read and results to analyze;

(6). Fig 3 shows the main hemodynamic results between COn and ECC. I would like to see possible reasons for these results in the discusison section

(7). Since individual responses were decribed in the section 3.4, why not provide a figure with these results for hemodynamics parameters? Maybe you can see more interesting results.

(8). Conclusion: delete line 331-332. Provide a more concise conclusion: what is the main difereence between Con e ECC for EMG and Hemodynamics parameters?

Reviewer 2 Report

The introduction is enriched by what is explained as expectations

 Lines nr 94-95: why were the power outputs and exercise modalities performed in a random order? 

Results: the data are shown in detail but not in schematic order. They are quite a lot. For this reason it is hard to discuss them.

Finally it is important to know if the participants were engaged in sports in childhood because their trainability could determine bias also because the sample size is small.

Conclusion are satisfying as the English language.

Regards

Author Response

#Reviewer 2

  1. The introduction is enriched by what is explained as expectations
  2. Lines nr 94-95: why were the power outputs and exercise modalities performed in a random order? 

It was so in order not to include the confound of an order effect in the analysis. Particularly, if exercise intensities had been performed in an increasing or decreasing order, participants would have adjusted their perception of effort based on this knowledge, and they may have reported what they thought they were supposed to feel rather than what they actually felt.

  1. Results: the data are shown in detail but not in schematic order. They are quite a lot. For this reason it is hard to discuss them.

We must agree. This is partly due to the use of non-parametric testing for perceptual responses, which do not put forth main effects; a Friedman’s ANOVA merely provides the authorization to use a post-hoc test. We split the results into more paragraph to help distinguish the details about each variable. We also added explanatory sentences to try to outline the main findings (lines 185, 160 to 163,  

As for the discussion we tried to remain as concise as possible to answer the main question of the study (i.e., difference in EMG and perceived effort before and after an extended familiarization period).

  1. Finally it is important to know if the participants were engaged in sports in childhood because their trainability could determine bias also because the sample size is small.

The participants were random healthy individuals, practicing physical activities as a leisure from a young age for most of them. This has now been specified (lines 68).

  1. Conclusion are satisfying as the English language.

Reviewer 3 Report

This paper investigated the impact of four short cycling sessions of submaximal ECC cycling on leg muscles EMG and perceived effort in individuals naive to the exercise. The results show that the activity of four leg muscles and perception of effort became lower in ECC than in CON cycling at most of five power outputs while they were similar before. The design of the experimental approach and procedure meets the professional standard, and the statistical data analysis demonstrates all the relevant factors that the readers care about in practical applications. In addition, the paper is well organized and written. Therefore, the reviewer recommends publication of this paper.

Author Response

Dear Reviewer, we thank you for your time and positive comments.

Round 2

Reviewer 1 Report

All modifications were provided

Author Response

All modifications were provided.

We thank you for having taken the time to assess our work a second time.

Reviewer 2 Report

Dear author, 

I'm plesased you welcomed my suggestions.
Here you can find a few more adjustments; however, your last version is definitively clearer than the previous one.

Lines 89-90: it's better to clearly state in the paragraph the reason why you preferred the random order. Having information about the sequence of exercises is essential for trainers as they affect the results. 

Lines 87-88: The recovery time between drills is essential too.
Keep in mind that your article will be read by exercise physiologists, trainers and other stakeholders, and recovery is a relevant part of a training session.

Lines 161-162. It is convenient to refer to the table right after mentioning it.

"Results" paragraph: still, too many variables are taken into account.
I know the article is supposed to be a deeper study compared to the less recent ones, but the parameters are too many.
Have you ever though to analyze just a shorter range of PPO? For example, just 30%, 75% and 90% PPO? or maybe to sort out the findings into structured and systematic tables?

Regards

Author Response

I'm plesased you welcomed my suggestions.
Here you can find a few more adjustments; however, your last version is definitively clearer than the previous one.

Lines 89-90: it's better to clearly state in the paragraph the reason why you preferred the random order. Having information about the sequence of exercises is essential for trainers as they affect the results. 

Reply: We specified line 91 that the random order served “to avoid the confound of an order effect”, though this is well known in research and justifications are usually asked only were the order is not random.

Lines 87-88: The recovery time between drills is essential too.
Keep in mind that your article will be read by exercise physiologists, trainers and other stakeholders, and recovery is a relevant part of a training session.

Reply: Indeed, this is something we should have specified earlier, please see lines 88 and 89.

Lines 161-162. It is convenient to refer to the table right after mentioning it.

"Results" paragraph: still, too many variables are taken into account.
I know the article is supposed to be a deeper study compared to the less recent ones, but the parameters are too many.
Have you ever though to analyze just a shorter range of PPO? For example, just 30%, 75% and 90% PPO? or maybe to sort out the findings into structured and systematic tables?

Reply: We welcome your suggestion of simplifying the results and thus only displayed the results for 45%, 60% and 90% PPO in the figures. However, we cannot hide the results in the manuscript as they were collected. In addition, removing the data of two intensities would not allow the computation of statistical difference in variation coefficient.